# Optimal Binary Classifier Aggregation for General Losses

**Akshay Balsubramani**
University of California, San Diego
abalsubr@ucsd.edu

**Yoav Freund**
University of California, San Diego
yfreund@ucsd.edu

## Abstract

We address the problem of aggregating an ensemble of predictors with known loss bounds in a semi-supervised binary classification setting, to minimize prediction loss incurred on the unlabeled data. We find the minimax optimal predictions for a very general class of loss functions including all convex and many non-convex losses, extending a recent analysis of the problem for misclassification error. The result is a family of semi-supervised ensemble aggregation algorithms which are as efficient as linear learning by convex optimization, but are minimax optimal without any relaxations. Their decision rules take a form familiar in decision theory – applying sigmoid functions to a notion of ensemble margin – without the assumptions typically made in margin-based learning.

## 1 Introduction

Consider a binary classification problem, in which we are given an ensemble of individual classifiers to aggregate into the most accurate predictor possible for data falling into two classes. Our predictions are measured on a large test set of unlabeled data, on which we know the ensemble classifiers' predictions but not the true test labels. Without using the unlabeled data, the prototypical supervised solution is empirical risk minimization (ERM): measure the errors of the ensemble classifiers with labeled data, and then simply predict according to the best classifier. But can we learn a better predictor by using unlabeled data as well?

This problem is central to semi-supervised learning. The authors of this paper recently derived the worst-case-optimal solution for it when performance is measured with classification error ([1]). However, this zero-one loss is inappropriate for other common binary classification tasks, such as estimating label probabilities, and handling false positives and false negatives differently. Such goals motivate the use of different evaluation losses like log loss and cost-weighted misclassification loss.

In this paper, we generalize the setup of [1] to these loss functions and a large class of others. Like the earlier work, the choice of loss function completely specifies the minimax optimal ensemble aggregation algorithm in our setting, which is efficient and scalable.

The algorithm learns weights over the ensemble classifiers by minimizing a convex function. The optimal prediction on each example in the test set is a sigmoid-like function of a linear combination of the ensemble predictions, using the learned weighting. Due to the minimax structure, this decision rule depends solely upon the loss function and upon the structure of the ensemble predictions on data, with no parameter or model choices.

### 1.1 Preliminaries

Our setting generalizes that of [1], in which we are given an ensemble $\mathcal{H} = \{h_1, \ldots, h_p\}$ and unlabeled (test) examples $x_1, \ldots, x_n$ on which to predict. The ensemble's predictions on the unlabeled

data are written as a matrix $\mathbf{F}$:

$$\mathbf{F} = \begin{pmatrix} h_1(x_1) & h_1(x_2) & \cdots & h_1(x_n) \\ \vdots & \vdots & \ddots & \vdots \\ h_p(x_1) & h_p(x_2) & \cdots & h_p(x_n) \end{pmatrix} \tag{1}$$

We use vector notation for the rows and columns of $\mathbf{F}$: $\mathbf{h}_i = (h_i(x_1), \cdots, h_i(x_n))^\top$ and $\mathbf{x}_j = (h_1(x_j), \cdots, h_p(x_j))^\top$. Each example $j \in [n]$ has a binary label $y_j \in \{-1, 1\}$, but the test labels are allowed to be randomized, represented by values in $[-1, 1]$ instead of just the two values $\{-1, 1\}$; e.g. $z_i = \frac{1}{2}$ indicates $y_i = +1$ w.p. $\frac{3}{4}$ and $-1$ w.p. $\frac{1}{4}$. So the labels on the test data can be represented by $\mathbf{z} = (z_1; \ldots; z_n) \in [-1, 1]^n$, and are unknown to the predictor, which predicts $\mathbf{g} = (g_1; \ldots; g_n) \in [-1, 1]^n$.

## 1.2 Loss Functions

We incur loss on test example $j$ according to its true label $y_j$. If $y_j = 1$, then the loss of predicting $g_j \in [-1, 1]$ on it is some function $\ell_+(g_j)$; and if $y_j = -1$, then the loss is $\ell_-(g_j)$. To illustrate, if the loss is the expected classification error, then $g_j \in [-1, 1]$ can be interpreted as a randomized binary prediction in the same way as $z_j$, so that $\ell_+(g_j) = \frac{1}{2}(1 - g_j)$ and $\ell_-(g_j) = \frac{1}{2}(1 + g_j)$.

We call $\ell_\pm$ the *partial losses* here, following earlier work (e.g. [16]). Since the true label can only be $\pm 1$, the partial losses fully specify the decision-theoretic problem we face, and changing them is tantamount to altering the prediction task.

What could such partial losses conceivably look like in general? Observe that they intuitively measure discrepancy to the true label $\pm 1$. Consequently, it is natural for e.g. $\ell_+(g)$ to be decreasing, as $g$ increases toward the notional true label $+1$. This suggests that both partial losses $\ell_+(\cdot)$ and $\ell_-(\cdot)$ would be monotonic, which we assume hereafter in this paper (throughout we use *increasing* to mean "monotonically nondecreasing" and vice versa).

**Assumption 1.** *Over the interval* $(-1, 1)$*,* $\ell_+(\cdot)$ *is decreasing and* $\ell_-(\cdot)$ *is increasing, and both are twice differentiable.*

We view Assumption 1 as very mild, as motivated above. Notably, convexity or symmetry of the partial losses are not required. In this paper, "general losses" refer to loss functions whose partial losses satisfy Assumption 1, to contrast them with convex losses or other subclasses.

The expected loss incurred w.r.t. the *randomized* true labels $z_j$ is a linear combination of the partial losses:

$$\ell(z_j, g_j) := \left(\frac{1 + z_j}{2}\right) \ell_+(g_j) + \left(\frac{1 - z_j}{2}\right) \ell_-(g_j) \tag{2}$$

Decision theory and learning theory have thoroughly investigated the nature of the loss $\ell$ and its partial losses, particularly how to estimate the "conditional label probability" $z_j$ using $\ell(z_j, g_j)$. A natural operation to do this is to minimize the loss over $g_j$, and a loss $\ell$ such that $\arg\min_{g \in [-1, 1]} \ell(z_j, g) = z_j$ (for all $z_j \in [-1, 1]$) is called a *proper loss* ([17, 16]).

## 1.3 Minimax Formulation

As in [1], we formulate the ensemble aggregation problem as a two-player zero-sum game between a predictor and an adversary. In this game, the first player is the predictor, playing predictions over the test set $\mathbf{g} \in [-1, 1]^n$. The adversary then sets the true labels $\mathbf{z} \in [-1, 1]^n$.

The key idea is that any ensemble constituent $i \in [p]$ known to have low loss on the test data gives us information about the unknown $\mathbf{z}$, as $\mathbf{z}$ is constrained to be "close" to the test predictions $\mathbf{h}_i$. Each hypothesis in the ensemble represents such a constraint, and $\mathbf{z}$ is in the intersection of all these constraint sets, which interact in ways that depend on the ensemble predictions $\mathbf{F}$.

Accordingly, for now assume the predictor knows a vector of label correlations $\mathbf{b}$ such that

$$\forall i \in [p]: \qquad \frac{1}{n} \sum_{j=1}^n h_i(x_j) z_j \geq b_i \tag{3}$$

i.e. $\frac{1}{n}\mathbf{Fz} \geq \mathbf{b}$. When the ensemble is composed of binary classifiers which predict in $[-1, 1]$, these $p$ inequalities represent upper bounds on individual classifier error rates. These can be estimated from the training set w.h.p. when the training and test data are i.i.d. using uniform convergence, exactly as in the prototypical supervised ERM procedure discussed in the introduction ([5]). So in our game-theoretic formulation, the adversary plays under ensemble constraints defined by $\mathbf{b}$.

The predictor's goal is to *minimize the worst-case expected loss of* $\mathbf{g}$ *on the test data* (w.r.t. the randomized labeling $\mathbf{z}$), using the loss function as defined earlier in Equation (2):

$$\ell(\mathbf{z}, \mathbf{g}) := \frac{1}{n} \sum_{j=1}^{n} \ell(z_j, g_j)$$

This goal can be written as the following optimization problem, a two-player zero-sum game:

$$V := \min_{\mathbf{g} \in [-1,1]^n} \max_{\substack{\mathbf{z} \in [-1,1]^n, \\ \frac{1}{n}\mathbf{Fz} \geq \mathbf{b}}} \ell(\mathbf{z}, \mathbf{g}) \tag{4}$$

$$= \min_{\mathbf{g} \in [-1,1]^n} \max_{\substack{\mathbf{z} \in [-1,1]^n, \\ \frac{1}{n}\mathbf{Fz} \geq \mathbf{b}}} \frac{1}{n} \sum_{j=1}^{n} \left[ \left( \frac{1+z_j}{2} \right) \ell_+(g_j) + \left( \frac{1-z_j}{2} \right) \ell_-(g_j) \right] \tag{5}$$

In this paper, we solve the learning problem faced by the predictor, finding an optimal strategy $\mathbf{g}^*$ realizing the minimum in (4) for any given "general loss" $\ell$. This strategy guarantees the best possible worst-case performance on the unlabeled dataset, with an upper bound of $V$ on the loss. Indeed, for all $\mathbf{z}_0$ and $\mathbf{g}_0$ obeying the constraints, Equation (4) implies the tight inequalities

$$\min_{\mathbf{g} \in [-1,1]^n} \ell(\mathbf{z}_0, \mathbf{g}) \overset{(a)}{\leq} V \leq \max_{\substack{\mathbf{z} \in [-1,1]^n, \\ \frac{1}{n}\mathbf{Fz} \geq \mathbf{b}}} \ell(\mathbf{z}, \mathbf{g}_0) \tag{6}$$

and $\mathbf{g}^*$ attains the equality in $(a)$, with a worst-case loss as good as *any* aggregated predictor.

In our formulation of the problem, the constraints on the adversary take a central role. As discussed in previous work with this formulation ([1, 2]), these constraints encode the information we have about the true labels. Without them, the adversary would find it optimal to trivially guarantee error (arbitrarily close to) $\frac{1}{2}$ by simply setting all labels uniformly at random ($\mathbf{z} = \mathbf{0}^n$). It is clear that adding information through more constraints will never raise the error bound $V$. [1]

Nothing has yet been assumed about $\ell(\mathbf{z}, \mathbf{g})$ other than Assumption 1. Our main results will require only this, holding for general losses. This brings us to this paper's contributions:

1. We give the exact minimax $\mathbf{g}^* \in [-1, 1]^n$ for general losses (Section 2.1). The optimal prediction on each example $j$ is a sigmoid function of a fixed linear combination of the ensemble's $p$ predictions on it, so $\mathbf{g}^*$ is a non-convex function of the ensemble predictions. By (6), this incurs the lowest worst-case loss of any predictor constructed with the ensemble information $\mathbf{F}$ and $\mathbf{b}$.

2. We derive an efficient algorithm for learning $\mathbf{g}^*$, by solving a $p$-dimensional convex optimization problem. This applies to a broad class of losses (cf. Lem. 2), including any with convex partial losses. Sec. 2 develops and discusses the results.

3. We extend the optimal $\mathbf{g}^*$ and efficient learning algorithm for it, as above, to a large variety of more general ensembles and prediction scenarios (Sec. 3), including constraints arising from general loss bounds, and ensembles of "specialists" and heterogeneous features.

## 2   Results for Binary Classification

Based on the loss, define the function $\Gamma : [-1, 1] \mapsto \mathbb{R}$ as $\Gamma(g) := \ell_-(g) - \ell_+(g)$. (We also write the vector $\Gamma(\mathbf{g})$ componentwise with $[\Gamma(\mathbf{g})]_j = \Gamma(g_j)$ for convenience, so that $\Gamma(\mathbf{h}_i) \in \mathbb{R}^n$ and $\Gamma(\mathbf{x}_j) \in \mathbb{R}^p$.) Observe that by Assumption 1, $\Gamma(g)$ is increasing on its domain; so we can discuss its inverse $\Gamma^{-1}(m)$, which is typically sigmoid-shaped, as will be illustrated.

With these we will set up the solution to the game (4), which relies on a convex function.

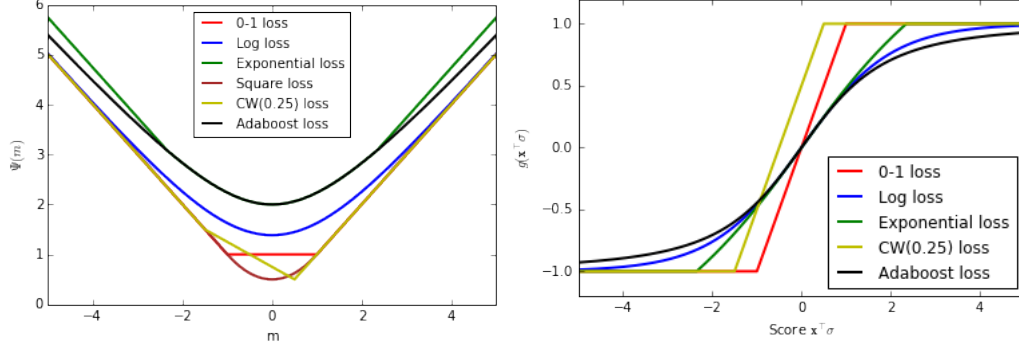

Figure 1: At left are plots of potential wells. At right are optimal prediction functions $g$, as a function of score. Both are shown for various losses, as listed in Section 2.3.

**Definition 1** (Potential Well). *Define the **potential well***

$$\Psi(m) := \begin{cases} -m + 2\ell_-(-1) & \text{if } m \leq \Gamma(-1) \\ \ell_+(\Gamma^{-1}(m)) + \ell_-(\Gamma^{-1}(m)) & \text{if } m \in (\Gamma(-1), \Gamma(1)) \\ m + 2\ell_+(1) & \text{if } m \geq \Gamma(1) \end{cases}$$

**Lemma 2.** *The potential well $\Psi(m)$ is continuous and 1-Lipschitz. It is also convex under* any *of the following conditions:*

    *(A) The partial losses $\ell_\pm(\cdot)$ are convex over $(-1, 1)$.*
    *(B) The loss function $\ell(\cdot, \cdot)$ is a proper loss.*
    *(C) $\ell'_-(x)\ell''_+(x) \geq \ell''_-(x)\ell'_+(x)$ for all $x \in (-1, 1)$.*

*Condition (C) is also necessary for convexity of $\Psi$, under Assumption 1.*

So the potential wells for different losses are shaped similarly, as seen in Figure 1. Lemma 2 tells us that the potential well is easy to optimize under any of the given conditions. Note that these conditions encompass convex surrogate losses commonly used in ERM, including all such "margin-based" losses (convex univariate functions of $z_j g_j$), introduced primarily for their favorable computational properties.

An easily optimized potential well benefits us, because the learning problem basically consists of optimizing it over the unlabeled data, as we will soon make explicit. The function that will actually be optimized is in terms of the dual parameters, so we call it the slack function.

**Definition 3** (Slack Function). *Let $\sigma \geq \mathbf{0}^p$ be a weight vector over $\mathcal{H}$ (not necessarily a distribution). The vector of **scores** is $\mathbf{F}^\top \sigma = (\mathbf{x}_1^\top \sigma, \ldots, \mathbf{x}_n^\top \sigma)$, whose elements' magnitudes are the **margins**. The prediction **slack function** is*

$$\gamma(\sigma, \mathbf{b}) := \gamma(\sigma) := -\mathbf{b}^\top \sigma + \frac{1}{n} \sum_{j=1}^{n} \Psi(\mathbf{x}_j^\top \sigma) \tag{7}$$

*An optimal weight vector $\sigma^*$ is any minimizer of the slack function: $\sigma^* \in \arg\min_{\sigma \geq \mathbf{0}^p} [\gamma(\sigma)]$.*

## 2.1 Solution of the Game

These are used to describe the minimax equilibrium of the game (4), in our main result.

**Theorem 4.** *The minimax value of the game (4) is*

$$\min_{\substack{\mathbf{g} \in [-1,1]^n}} \max_{\substack{\mathbf{z} \in [-1,1]^n, \\ \frac{1}{n}\mathbf{F}\mathbf{z} \geq \mathbf{b}}} \ell(\mathbf{z}, \mathbf{g}) = V = \frac{1}{2}\gamma(\sigma^*) = \frac{1}{2} \min_{\sigma \geq \mathbf{0}^p} \left[ -\mathbf{b}^\top \sigma + \frac{1}{n} \sum_{j=1}^{n} \Psi(\mathbf{x}_j^\top \sigma) \right]$$

*The minimax optimal predictions are defined as follows: for all $j \in [n]$,*

$$g_j^* := g_j(\sigma^*) = \begin{cases} -1 & \text{if} \quad \mathbf{x}_j^\top \sigma^* \leq \Gamma(-1) \\ \Gamma^{-1}(\mathbf{x}_j^\top \sigma^*) & \text{if} \quad \mathbf{x}_j^\top \sigma^* \in (\Gamma(-1), \Gamma(1)) \\ 1 & \text{if} \quad \mathbf{x}_j^\top \sigma^* \geq \Gamma(1) \end{cases} \tag{8}$$

$g_j^*$ is always an increasing sigmoid, as shown in Figure 1.

We can also redo the proof of Theorem 4 when $\mathbf{g} \in [-1, 1]^n$ is not left as a free variable set in the game, but instead is preset to $\mathbf{g}(\sigma)$ as in (8) for some (possibly suboptimal) weight vector $\sigma$.

**Observation 5.** *For any weight vector $\sigma_0 \geq \mathbf{0}^p$, the worst-case loss after playing $\mathbf{g}(\sigma_0)$ is*

$$\max_{\substack{\mathbf{z} \in [-1,1]^n, \\ \frac{1}{n}\mathbf{Fz} \geq \mathbf{b}}} \ell(\mathbf{z}, \mathbf{g}(\sigma_0)) \leq \frac{1}{2}\gamma(\sigma_0)$$

The proof is a simplified version of that of Theorem 4; there is no minimum over $\mathbf{g}$ to deal with, and the minimum over $\sigma \geq \mathbf{0}^p$ in Equation (13) is upper-bounded by using $\sigma_0$. This result is an expression of weak duality in our setting, and generalizes Observation 4 of [1].

## 2.2 Ensemble Aggregation Algorithm

Theorem 4 defines a prescription for aggregating the given ensemble predictions on the test set.

**Learning:** *Minimize the slack function $\gamma(\sigma)$, finding the minimizer $\sigma^*$ that achieves $V$.*
This is a convex optimization under broad conditions (Lemma 2), and when the test examples are i.i.d. the $\Psi$ term is a sum of $n$ i.i.d. functions. Therefore, it is readily amenable to standard first-order optimization methods which require only $O(1)$ test examples at once. In practice, learning employs such methods to *approximately* minimize $\gamma$, finding some $\sigma_A$ such that $\gamma(\sigma_A) \leq \gamma(\sigma^*) + \epsilon$ for some small $\epsilon$. Standard convex optimization methods are guaranteed to do this for binary classifier ensembles, because the slack function is Lipschitz (Lemma 2) and $\|\mathbf{b}\|_\infty \leq 1$.

**Prediction:** *Predict $g(\sigma^*)$ on any test example, as indicated in (8).*
This decouples the prediction task over each test example separately, which requires $O(p)$ time and memory like $p$-dimensional linear prediction. After finding an $\epsilon$-approximate minimizer $\sigma_A$ in the learning step as above, Observation 5 tells us that the prediction $\mathbf{g}(\sigma_A)$ has loss $\leq V + \frac{\epsilon}{2}$.

In particular, note that there is no algorithmic dependence on $n$ in either step in a statistical learning setting. So though our formulation is transductive, it is no less tractable than a stochastic optimization setting in which i.i.d. data arrive one at a time, and applies to this common situation.

## 2.3 Examples of Different Losses

To further illuminate Theorem 4, we detail a few special cases in which $\ell_\pm$ are explicitly defined. These losses may be found throughout the literature (see e.g. [16]). The key functions $\Psi$ and $g^*$ are listed for these losses in Appendix A, and in many cases in Figure 1. The nonlinearities used for $g^*$ are sigmoids, arising solely from the intrinsic minimax structure of the classification game.

- **0-1 Loss**: Here $g_j$ is taken to be a randomized binary prediction; this case was developed in [1], the work we generalize in this paper.
- **Log Loss**, **Square Loss**
- **Cost-Weighted Misclassification (Quantile) Loss**: This is defined with a parameter $c \in [0, 1]$ representing the relative cost of false positives vs. false negatives, making the Bayes-optimal classifier the $c$-quantile of the conditional probability distribution ([19]).
- **Exponential Loss**, **Logistic Loss**
- **Hellinger Loss**: This is typically given for $p, y \in [0, 1]$ as $\frac{1}{2}\left(\left(\sqrt{p} - \sqrt{y}\right)^2 + \left(\sqrt{1-p} - \sqrt{1-y}\right)^2\right)$. Our formulation is equivalent when the prediction and label are rescaled to $[-1, 1]$.

- **"AdaBoost Loss"**: If the goal of AdaBoost ([18]) is interpreted as class probability estimation, the implied loss is proper and given in [6, 16].
- **Absolute Loss** and **Hinge Loss**: The absolute loss can be defined by $\ell_{\mp}^{abs}(g_j) = 1 \pm g_j$, and the hinge loss also has $\ell_{\mp}^{abs}(g_j) = 1 \pm g_j$ since the kink in the hinge loss only lies at $g_j = \mp 1$. These partial losses are the same as for 0-1 loss up to scaling, and therefore all our results for $\Psi$ and $\mathbf{g}^*$ are as well. So these losses are not shown in Appendix A.
- **Sigmoid Loss**: This is an example of a sigmoid-shaped margin loss, a nonconvex smooth surrogate for 0-1 loss. Similar losses have arisen in a variety of binary classification contexts, from robustness (e.g. [9]) to active learning ([10]) and structured prediction ([14]).

## 2.4 Related Work and Technical Discussion

There are two notable ways in which the result of Theorem 4 is particularly advantageous and general. First, the fact that $\ell(z, g)$ can be non-convex in $g$, yet solvable by convex optimization, is a major departure from previous work. Second, the solution has a convenient dependence on $n$ (as in [1]), simply averaging a function over the unlabeled data, which is not only mathematically convenient but also makes stochastic $O(1)$-space optimization practical. This is surprisingly powerful, because the original minimax problem is *jointly* over the entire dataset, avoiding further independence or decoupling assumptions.

Both these favorable properties stem from the structure of the binary classification problem, as we can describe by examining the optimization problem constructed within the proof of Thm. 4 (Appendix C.1). In it, the constraints which do not explicitly appear with Lagrange parameters are all box, or $L_\infty$ norm, constraints. These decouple over the $n$ test examples, so the problem can be reduced to the one-dimensional optimization at the heart of Eq. (14), which is solved ad hoc. So we are able to obtain minimax results for these non-convex problems – the $g_i$ are "clipped" sigmoid functions because of the bounding effect of the $[-1, 1]$ box constraints intrinsic to binary classification. We introduce Lagrange parameters $\sigma$ only for the $p$ remaining constraints in the problem, which do not decouple as above, applying globally over the $n$ test examples. However, these constraints only depend on $n$ as an average over examples (which is how they arise in dual form in Equation (16) of the proof), and the loss function itself is also an average (Equation (12)). This makes the box constraint decoupling possible, and leads to the favorable dependence on $n$, making an efficient solution possible to a potentially flagrantly non-convex problem.

To summarize, the technique of optimizing only "halfway into" the dual allows us to readily manipulate the minimax problem exactly without using an approximation like weak duality, despite the lack of convexity in $\mathbf{g}$. This technique was used implicitly for a different purpose in the "drifting game" analysis of boosting ([18], Sec. 13.4.1). Existing boosting work is loosely related to our approach in being a transductive game invoked to analyze ensemble aggregation, but it does not consider unlabeled data and draws its power instead from being a repeated game ([18]).

The predecessor to this work ([1]) addresses a problem, 0-1 loss minimization, that is known to be NP-hard when solved directly ([11]). Using the unlabeled data is essential to surmounting this. It gives the dual problem an independently interesting interpretation, so the learning problem is on the always-convex Lagrange dual function and is therefore tractable.

This paper's transductive formulation involves no surrogates or relaxations of the loss, in sharp contrast to most previous work. This allows us to bypass the consistency and agnostic-learning discussions ([22, 3]) common to ERM methods that use convex risk minimization. Convergence analyses of those methods make heavy use of convexity of the losses and are generally done presupposing a linear weighting over $\mathcal{H}$ ([21]), whereas here such structure emerges directly from Lagrange duality and involves no convexity to derive the worst-case-optimal predictions.

The conditions in Assumption 1 are notably general. Differentiability of the partial losses is convenient, but not necessary, and only used because first-order conditions are a convenient way to establish convexity of the potential well in Lemma 2. It is never used elsewhere, including in the minimax arguments used to prove Theorem 4. These manipulations are structured to be valid even if $\ell_\pm$ are non-monotonic; but in this case, $g_j^*$ could turn out to be multi-valued and hence not a genuine function of the example's score.

We emphasize that our result on the minimax equilibrium (Theorem 4) holds for general losses – the slack function may not be convex unless the further conditions of Lemma 2 are met, but

the interpretation of the optimum in terms of margins and sigmoid functions remains. All this emerges from the inherent decision-theoretic structure of the problem (the proof of Appendix C.1). It manifests in the fact that the function $g(\mathbf{x}_j^\top \sigma)$ is always increasing in $\mathbf{x}_j^\top \sigma$ for general losses, because the function $\Gamma$ is increasing. This monotonicity typically needs to be assumed in a generalized linear model (GLM; [15]) and related settings. $\Gamma$ appears loosely analogous to the link function used by GLMs, with its inverse being used for prediction.

The optimal decision rules emerging from our framework are *artificial neurons* of the ensemble inputs. Helmbold et al. introduce the notion of a "matching loss" ([13]) for learning the parameters of a (fully supervised) artificial neuron with an arbitrary increasing transfer function, effectively taking the opposite tack of this paper in using a neuron's transfer function to derive a loss to minimize in order to learn the neuron's weights by convex optimization. Our assumptions on the loss, particularly condition (C) of Lemma 2, have arisen independently in earlier online learning work by some of the same authors ([12]); this may suggest connections between our techniques. We also note that our family of general losses was defined independently by [19] in the ERM setting (dubbed "F-losses") – in which condition (C) of Lemma 2 also has significance ([19], Prop. 2) – but has seemingly not been revisited thereafter. Further fleshing out the above connections would be interesting future work.

# 3 Extensions

We detail a number of generalizations to the basic prediction scenario of Sec. 2. These extensions are not mutually exclusive, and apply in conjunction with each other, but we list them separately for clarity. They illustrate the versatility of our minimax framework, particularly Sec. 3.4.

## 3.1 Weighted Test Sets and Covariate Shift

Though our results here deal with binary classification of an unweighted test set, the formulation deals with a weighted set with only a slight modification of the slack function:

**Theorem 6.** *For any vector* $\mathbf{r} \geq \mathbf{0}^n$,

$$
\min_{\mathbf{g} \in [-1,1]^n} \max_{\substack{\mathbf{z} \in [-1,1]^n, \\ \frac{1}{n}\mathbf{Fz} \geq \mathbf{b}}} \frac{1}{n} \sum_{j=1}^n r_j \ell(z_j, g_j) = \frac{1}{2} \min_{\sigma \geq \mathbf{0}^p} \left[ -\mathbf{b}^\top \sigma + \frac{1}{n} \sum_{j=1}^n r_j \Psi\left( \frac{\mathbf{x}_j^\top \sigma}{r_j} \right) \right]
$$

*Writing* $\sigma_{\mathbf{r}}^*$ *as the minimizer of the RHS above, the optimal predictions* $\mathbf{g}^* = \mathbf{g}(\sigma_{\mathbf{r}}^*)$, *as in Theorem 4.*

Such weighted classification can be parlayed into algorithms for general supervised learning problems via learning reductions ([4]). Allowing weights on the test set for the evaluation is tantamount to accounting for known covariate shift in our setting; it would be interesting, though outside our scope, to investigate scenarios with more uncertainty.

## 3.2 General Loss Constraints on the Ensemble

So far in the paper, we have considered the constraints on ensemble classifiers as derived from their label correlations (i.e. 0-1 losses), as in (3). However, this view can be extended significantly with the same analysis, because any general loss $\ell(z, g)$ is linear in $z$ (Eq. (2)), which allows our development to go through essentially intact.

In summary, a classifier can be incorporated into our framework for aggregation if we have a generalization loss bound on it, for any "general loss." This permits an enormous variety of constraint sets, as each classifier considered can have constraints corresponding to any number of loss bounds on it, even multiple loss bounds using different losses. For instance, $h_1$ can yield a constraint corresponding to a zero-one loss bound, $h_2$ can yield one constraint corresponding to a square loss bound and another corresponding to a zero-one loss bound, and so on. Appendix B details this idea, extending Theorem 4 to general loss constraints.

## 3.3 Uniform Convergence Bounds for $\mathbf{b}$

In our basic setup, $\mathbf{b}$ has been taken as a lower bound on ensemble classifier label correlations. But as mentioned earlier, the error in estimating $\mathbf{b}$ is in fact often quantified by two-sided uniform

convergence ($L_\infty$) bounds on $\mathbf{b}$. Constraining $\mathbf{z}$ in this fashion amounts to $L_1$ regularization of the dual vector $\sigma$.

**Proposition 7.** *For any $\epsilon \geq 0$,*

$$\min_{\mathbf{g} \in [-1,1]^n} \max_{\substack{\mathbf{z} \in [-1,1]^n, \\ \left\| \frac{1}{n}\mathbf{Fz}-\mathbf{b} \right\|_\infty \leq \epsilon}} \ell(\mathbf{z}, \mathbf{g}) \;=\; \min_{\sigma \in \mathbb{R}^p} \left[ -\mathbf{b}^\top \sigma + \frac{1}{n}\sum_{j=1}^n \Psi(\mathbf{x}_j^\top \sigma) + \epsilon \left\| \sigma \right\|_1 \right]$$

*As in Thm. 4, the optimal $\mathbf{g}^* = \mathbf{g}(\sigma_\infty^*)$, where $\sigma_\infty^*$ is the minimizer of the right-hand side above.*

Here we optimize over all vectors $\sigma$ (not just nonnegative ones) in an $L_1$-regularized problem, convenient in practice because we can cross-validate over the parameter $\epsilon$. To our knowledge, this particular analysis has been addressed in prior work only for the special case of the entropy loss on the probability simplex, discussed further in [8].

Prop. 7 is a corollary of a more general result using differently scaled label correlation deviations within the ensemble, i.e. $\left| \frac{1}{n}\mathbf{Fz} - \mathbf{b} \right| \leq \mathbf{c}$ for a general $\mathbf{c} \geq \mathbf{0}^n$. This turns out to be equivalent to regularizing the minimization over $\sigma$ by its $\mathbf{c}$-weighted $L_1$ norm $\mathbf{c}^\top |\sigma|$ (Thm. 11), used to penalize the ensemble nonuniformly ([7]). This situation is motivated by uniform convergence of heterogeneous ensembles comprised of e.g. "specialist" predictors, for which a union bound ([5]) results in $\left| \frac{1}{n}\mathbf{Fz} - \mathbf{b} \right|$ with varying coordinates. Such ensembles are discussed next.

### 3.4 Heterogenous Ensembles of Specialist Classifiers and Features

All the results and algorithms in this paper apply in full generality to ensembles of "specialist" classifiers that only predict on some subset of the test examples. This is done by merely calculating the constraints over only these examples, and changing $\mathbf{F}$ and $\mathbf{b}$ accordingly ([2]).

To summarize this from [2], suppose a classifier $i \in [p]$ decides to abstain on an example $x_j$ ($j \in [n]$) with probability $1 - v_i(x)$, and otherwise predict $h_i(x)$. Our only assumption on $\{v_i(x_1), \ldots, v_i(x_n)\}$ is the reasonable one that $\sum_{j=1}^n v_i(x_j) > 0$, so classifier $i$ is not a useless specialist that abstains everywhere.

The information about $\mathbf{z}$ contributed by classifier $i$ is now not its overall correlation with $\mathbf{z}$ on the entire test set, but rather the correlation with $\mathbf{z}$ restricted to the test examples on which it predicts. On the test set, this is written as $\frac{1}{n}\mathbf{Sz}$, where the matrix $\mathbf{S}$ is formed by reweighting each row of $\mathbf{F}$:

$$\mathbf{S} := n \begin{pmatrix} \rho_1(x_1)h_1(x_1) & \rho_1(x_2)h_1(x_2) & \cdots & \rho_1(x_n)h_1(x_n) \\ \rho_2(x_1)h_2(x_1) & \rho_2(x_2)h_2(x_2) & \cdots & \rho_2(x_n)h_2(x_n) \\ \vdots & \vdots & \ddots & \vdots \\ \rho_p(x_1)h_p(x_1) & \rho_p(x_2)h_p(x_2) & \cdots & \rho_p(x_n)h_p(x_n) \end{pmatrix} \quad, \quad \rho_i(x_j) := \frac{v_i(x_j)}{\sum_{k=1}^n v_i(x_k)}$$

($\mathbf{S} = \mathbf{F}$ when the entire ensemble consists of non-specialists, recovering our initial setup.) Therefore, the ensemble constraints (3) become $\frac{1}{n}\mathbf{Sz} \geq \mathbf{b}_S$, where $\mathbf{b}_S$ gives the label correlations of each classifier restricted to the examples on which it predicts. Though this rescaling results in entries of $\mathbf{S}$ having different ranges and magnitudes $\geq 1$, our results and proofs remain entirely intact.

Indeed, despite the title, this paper applies far more generally than to an ensemble of binary classifiers, because our proofs make no assumptions at all about the structure of $\mathbf{F}$. Each predictor in the ensemble can be thought of as a feature; it has so far been convenient to think of it as binary, following the perspective of binary classifier aggregation, but it could as well be e.g. real-valued, and the features can have very different scales (as in $\mathbf{S}$ above). An unlabeled example $\mathbf{x}$ is simply a vector of features, so arbitrarily abstaining specialists are equivalent to "missing features," which this framework handles seamlessly due to the given unlabeled data. Our development *applies generally to semi-supervised binary classification*.

### Acknowledgements

AB is grateful to Chris "Ceej" Tosh for feedback that made the manuscript clearer. This work was supported by the NSF (grant IIS-1162581).

## Footnotes

[1] However, it may pose difficulties in estimating $\mathbf{b}$ by applying uniform convergence over a larger $\mathcal{H}$ ([2]).
