[Supplementary Material]

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

# A  Examples of Optimal Decision Rules for Various Losses

| Loss | Partial Losses | $\Gamma(g)$ | $\Psi(m)$ | $g_i(\sigma)$ |
|---|---|---|---|---|
| 0-1 | $\ell_-(g) = \frac{1}{2}(1+g)$ <br> $\ell_+(g) = \frac{1}{2}(1-g)$ | $g$ | $\max(1, |m|)$ | $\text{clip}(\mathbf{x}_i^\top \sigma)$ |
| Log | $\ell_-(g) = \ln\left(\frac{2}{1-g}\right)$ <br> $\ell_+(g) = \ln\left(\frac{2}{1+g}\right)$ | $\ln\left(\frac{1+g}{1-g}\right)$ | $\ln(1+e^m) + \ln(1+e^{-m})$ | $\dfrac{1 - e^{-\mathbf{x}_i^\top \sigma}}{1 + e^{-\mathbf{x}_i^\top \sigma}}$ |
| Square | $\ell_-(g) = \left(\frac{1+g}{2}\right)^2$ <br> $\ell_+(g) = \left(\frac{1-g}{2}\right)^2$ | $g$ | $\begin{cases} -m & m \le -1 \\ \frac{1}{2}(m^2+1) & m \in (-1,1) \\ m & m \ge 1 \end{cases}$ | $\text{clip}(\mathbf{x}_i^\top \sigma)$ |
| CW (param. c) | $\ell_-(g) = c(1+g)$ <br> $\ell_+(g) = (1-c)(1-g)$ | $g + 2c - 1$ | $\begin{cases} -m & m \le 2c-2 \\ (2c-1)m + 4c(1-c) & m \in (2c-2, 2c) \\ m & m \ge 2c \end{cases}$ | $\text{clip}(\mathbf{x}_i^\top \sigma + 1 - 2c)$ |
| Exponential | $\ell_-(g) = e^g$ <br> $\ell_+(g) = e^{-g}$ | $e^g - e^{-g}$ | $\begin{cases} -m + 2/e & m \le -e + \frac{1}{e} \\ \sqrt{4+m^2} & m \in (-e+\frac{1}{e}, e-\frac{1}{e}) \\ m + 2/e & m \ge e - \frac{1}{e} \end{cases}$ | $\text{clip}\left(\ln\left(\frac{1}{2}\mathbf{x}_i^\top \sigma + \sqrt{1 + \frac{1}{4}(\mathbf{x}_i^\top \sigma)^2}\right)\right)$ |
| Logistic | $\ell_-(g) = \ln(1+e^g)$ <br> $\ell_+(g) = \ln(1+e^{-g})$ | $g$ | $\begin{cases} -m + 2\ln(1+1/e) & m \le -1 \\ \ln(1+e^m) + \ln(1+e^{-m}) & m \in (-1,1) \\ m + 2\ln(1+1/e) & m \ge 1 \end{cases}$ | $\text{clip}(\mathbf{x}_i^\top \sigma)$ |
| Hellinger | $\ell_-(g) = 1 - \sqrt{\frac{1-g}{2}}$ <br> $\ell_+(g) = 1 - \sqrt{\frac{1+g}{2}}$ | $\sqrt{\frac{1+g}{2}} - \sqrt{\frac{1-g}{2}}$ | $\begin{cases} -m & m \le -1 \\ 2 - \sqrt{\frac{1-m\sqrt{2-m^2}}{2}} - \sqrt{\frac{1+m\sqrt{2-m^2}}{2}} & m \in (-1,1) \\ m & m \ge 1 \end{cases}$ | $\begin{cases} (\mathbf{x}_i^\top \sigma)\sqrt{2 - (\mathbf{x}_i^\top \sigma)^2} & |\mathbf{x}_i^\top \sigma| \le 1 \\ \text{sgn}(\mathbf{x}_i^\top \sigma) & |\mathbf{x}_i^\top \sigma| > 1 \end{cases}$ |
| "AdaBoost" | $\ell_-(g) = \sqrt{\frac{1+g}{1-g}}$ <br> $\ell_+(g) = \sqrt{\frac{1-g}{1+g}}$ | $\frac{2g}{\sqrt{1-g^2}}$ | $\sqrt{\frac{\sqrt{m^2+4}+m}{\sqrt{m^2+4}-m}} + \sqrt{\frac{\sqrt{m^2+4}-m}{\sqrt{m^2+4}+m}}$ | $\dfrac{\mathbf{x}_i^\top \sigma}{\sqrt{(\mathbf{x}_i^\top \sigma)^2 + 4}}$ |
| Sigmoid | $\ell_-(g) = \frac{1}{1+e^{-g}}$ <br> $\ell_+(g) = \frac{1}{1+e^g}$ | $\frac{e^g - 1}{e^g + 1}$ | $\max\left(1, |m| + \frac{2}{1+e}\right)$ | $\text{clip}\left(\ln\left(\frac{1+m}{1-m}\right)\right)$ |

Table 1: Some binary classification losses, as in Sec. 2.3. For convenience, we write $\text{clip}(x) = \min(1, \max(-1, x))$.

# B Constraints on General Losses for Binary Classification

In all other sections of the paper, we allow the evaluation function of the game to be a general loss, but assume that the constraints (our information about the ensemble) are in terms of zero-one loss as written in (3). However, here we relax that assumption, allowing each classifier $h_i$ to constrain the test labels $\mathbf{z}$ not with the zero-one loss of $h_i$'s predictions, but rather with some other general loss.

This is possible because when the true labels are binary, all the losses we consider are linear in $\mathbf{z}$, as seen in (5): $\ell(\mathbf{z}, \mathbf{g}) = \frac{1}{n} \sum_{j=1}^{n} \frac{1}{2} [\ell_+(g_j) + \ell_-(g_j)] - \frac{1}{2n} \mathbf{z}^\top [\Gamma(\mathbf{g})]$. Accordingly, recall that $\mathbf{h}_i \in [-1, 1]^n$ is the vector of test predictions of classifier $h_i$. Suppose we have an upper bound on the generalization loss of $h_i$, i.e. $\ell(\mathbf{z}, \mathbf{h}_i) \leq \epsilon_i^\ell$. If we define $b_i^\ell := \frac{1}{n} \sum_{j=1}^{n} [\ell_+(h_i(x_j)) + \ell_-(h_i(x_j))] - 2\epsilon_i^\ell$, then we can use the definition of $\ell(\mathbf{z}, \mathbf{g})$ to write

$$\ell(\mathbf{z}, \mathbf{h}_i) \leq \epsilon_i^\ell \qquad \Longleftrightarrow \qquad \frac{1}{n} \mathbf{z}^\top [\Gamma(\mathbf{h}_i)] \geq b_i^\ell \qquad (9)$$

Now (9) is a linear constraint on $\mathbf{z}$, just like each of the error constraints earlier considered in (3). We can derive an aggregation algorithm with constraints like (9), using essentially the same analysis as employed in Section 2 to solve the game (4). As mentioned in Sec. 3.2,

## Matching Objective and Constraint Losses

Though the ensemble constraints can be completely heterogeneous, we focus on a special case of them in the rest of this section to glean intuition. Suppose when each classifier contributes exactly one constraint to the problem, and the losses used for these constraints are all the same as each other and as the loss $\ell$ used in the objective function. In other words, the minimax prediction problem we now consider is

$$V^\ell := \min_{\substack{\mathbf{g} \in [-1,1]^n}} \max_{\substack{\mathbf{z} \in [-1,1]^n, \\ \forall i \in [p]: \, \ell(\mathbf{z}, \mathbf{h}_i) \leq \epsilon_i^\ell}} \ell(\mathbf{z}, \mathbf{g}) = \min_{\substack{\mathbf{g} \in [-1,1]^n}} \max_{\substack{\mathbf{z} \in [-1,1]^n, \\ \forall i \in [p]: \, \frac{1}{n} \mathbf{z}^\top [\Gamma(\mathbf{h}_i)] \geq b_i^\ell}} \ell(\mathbf{z}, \mathbf{g}) \qquad (10)$$

The matrix $\mathbf{F}$ and the slack function from (1) are therefore redefined:

$$F_{ij}^\ell := \Gamma(h_i(x_j)) = \ell_-(h_i(x_j)) - \ell_+(h_i(x_j))$$

$$\gamma^\ell(\sigma, \mathbf{b}^\ell) := \gamma^\ell(\sigma) := -[\mathbf{b}^\ell]^\top \sigma + \frac{1}{n} \sum_{j=1}^{n} \Psi \left( [\Gamma(\mathbf{x}_j)]^\top \sigma \right)$$

where $\mathbf{b}^\ell = (b_1^\ell, \ldots, b_p^\ell)^\top$ and the vectors $\mathbf{x}_j$ are now from the new unlabeled data matrix $F_{ij}^\ell$. The game (10) is clearly of the same form as the earlier formulation (4). Therefore, its solution has the same structure as in Theorem 4, proved using that theorem's proof:

**Theorem 8.** *The minimax value of the game* (10) *is* $V := \frac{1}{2} \gamma^\ell(\sigma^{\ell*}) := \min_{\sigma \geq \mathbf{0}^p} \frac{1}{2} \gamma^\ell(\sigma)$. *The minimax optimal predictions are defined as follows: for all* $j \in [n]$,

$$g_j^* := g_j(\sigma^*) = \begin{cases} -1 & \text{if } [\Gamma(\mathbf{x}_j)]^\top \sigma^{\ell*} \leq \Gamma(-1) \\ \Gamma^{-1} \left( [\Gamma(\mathbf{x}_j)]^\top \sigma^{\ell*} \right) & \text{if } [\Gamma(\mathbf{x}_j)]^\top \sigma^{\ell*} \in (\Gamma(-1), \Gamma(1)) \\ 1 & \text{if } [\Gamma(\mathbf{x}_j)]^\top \sigma^{\ell*} \geq \Gamma(1) \end{cases}$$

This provides a concise characterization of how to solve the semi-supervised binary classification game for general losses. Though on the face of it Theorem 8 is a much stronger result than even Theorem 4, we cannot overlook statistical issues. The loss bounds $\epsilon_i^\ell$ on each classifier are estimated using a uniform convergence bound over the ensemble with loss $\ell$, but the data now considered are not the ensemble predictions, but the predictions passed through function $\Gamma$. This can be impractical for losses like log loss, for which $\Gamma$ is unbounded, and therefore uniform convergence to estimate $b_i^\ell$ in (9) is much less applicable than for 0-1 loss.

But such issues are outside our scope here, and our constrained minimax results hold in any case given $\mathbf{b}$. They may be useful to obtain semi-supervised learnability results for different losses from tighter statistical characterizations, which we consider an interesting open problem.

# C Proofs and Supporting Results

## C.1 Proof of Theorem 4

The main hurdle here is the constrained maximization over $\mathbf{z}$. For this we use the following result, a basic application of Lagrange duality (from [1], but proved below for completeness).

**Lemma 9.** *For any* $\mathbf{a} \in \mathbb{R}^n$,

$$\max_{\substack{\mathbf{z} \in [-1,1]^n, \\ \frac{1}{n}\mathbf{F}\mathbf{z} \geq \mathbf{b}}} \frac{1}{n}\mathbf{z}^\top \mathbf{a} = \min_{\sigma \geq \mathbf{0}^p} \left[ -\mathbf{b}^\top \sigma + \frac{1}{n} \left\| \mathbf{F}^\top \sigma + \mathbf{a} \right\|_1 \right]$$

With this lemma, we prove the main theorem of this paper.

*Proof of Theorem 4.* First note that $\ell(\mathbf{z}, \mathbf{g})$ is linear in $\mathbf{z}$,

$$V = (5) = \frac{1}{2} \min_{\mathbf{g} \in [-1,1]^n} \max_{\substack{\mathbf{z} \in [-1,1]^n, \\ \frac{1}{n}\mathbf{F}\mathbf{z} \geq \mathbf{b}}} \frac{1}{n} \sum_{j=1}^{n} \left[ \ell_+(g_j) + \ell_-(g_j) + z_j \left( \ell_+(g_j) - \ell_-(g_j) \right) \right]$$

Here we can rewrite the constrained maximization over $\mathbf{z}$ using Lemma 9:

$$\max_{\substack{\mathbf{z} \in [-1,1]^n, \\ \frac{1}{n}\mathbf{F}\mathbf{z} \geq \mathbf{b}}} \frac{1}{n} \sum_{i=1}^{n} z_j \left( \ell_+(g_j) - \ell_-(g_j) \right) = \max_{\substack{\mathbf{z} \in [-1,1]^n, \\ \frac{1}{n}\mathbf{F}\mathbf{z} \geq \mathbf{b}}} -\frac{1}{n}\mathbf{z}^\top [\Gamma(\mathbf{g})]$$

$$= \min_{\sigma \geq \mathbf{0}^p} \left[ -\mathbf{b}^\top \sigma + \frac{1}{n} \left\| \mathbf{F}^\top \sigma - \Gamma(\mathbf{g}) \right\|_1 \right] \qquad (11)$$

Substituting (11) into (5) and simplifying,

$$V = \frac{1}{2} \min_{\mathbf{g} \in [-1,1]^n} \left[ \frac{1}{n} \sum_{j=1}^{n} \left[ \ell_+(g_j) + \ell_-(g_j) \right] + \max_{\substack{\mathbf{z} \in [-1,1]^n, \\ \frac{1}{n}\mathbf{F}\mathbf{z} \geq \mathbf{b}}} \frac{1}{n} \sum_{j=1}^{n} z_j \left( \ell_+(g_j) - \ell_-(g_j) \right) \right] \qquad (12)$$

$$= \frac{1}{2} \min_{\mathbf{g} \in [-1,1]^n} \left[ \frac{1}{n} \sum_{j=1}^{n} \left[ \ell_+(g_j) + \ell_-(g_j) \right] + \min_{\sigma \geq \mathbf{0}^p} \left[ -\mathbf{b}^\top \sigma + \frac{1}{n} \left\| \mathbf{F}^\top \sigma - \Gamma(\mathbf{g}) \right\|_1 \right] \right]$$

$$= \frac{1}{2} \min_{\sigma \geq \mathbf{0}^p} \left[ -\mathbf{b}^\top \sigma + \min_{\mathbf{g} \in [-1,1]^n} \left[ \frac{1}{n} \sum_{j=1}^{n} \left[ \ell_+(g_j) + \ell_-(g_j) \right] + \frac{1}{n} \left\| \mathbf{F}^\top \sigma - \Gamma(\mathbf{g}) \right\|_1 \right] \right] \qquad (13)$$

$$= \frac{1}{2} \min_{\sigma \geq \mathbf{0}^p} \left[ -\mathbf{b}^\top \sigma + \frac{1}{n} \sum_{j=1}^{n} \min_{g_j \in [-1,1]} \left[ \ell_+(g_j) + \ell_-(g_j) + \left| \mathbf{x}_j^\top \sigma - \Gamma(g_j) \right| \right] \right] \qquad (14)$$

The absolute value breaks down into two cases, so the inner minimization's objective can be simplified:

$$\ell_+(g_j) + \ell_-(g_j) + \left| \mathbf{x}_j^\top \sigma - \Gamma(g_j) \right| = \begin{cases} 2\ell_+(g_j) + \mathbf{x}_j^\top \sigma & \text{if } \mathbf{x}_j^\top \sigma \geq \Gamma(g_j) \\ 2\ell_-(g_j) - \mathbf{x}_j^\top \sigma & \text{if } \mathbf{x}_j^\top \sigma < \Gamma(g_j) \end{cases} \qquad (15)$$

Suppose $g_j$ falls in the first case, so that $\mathbf{x}_j^\top \sigma \geq \Gamma(g_j)$. From Assumption 1, $2\ell_+(g_j) + \mathbf{x}_j^\top \sigma$ is decreasing in $g_j$, so it is minimized for the greatest $g_j^* \leq 1$ s.t. $\Gamma(g_j^*) \leq \mathbf{x}_j^\top \sigma$. Since $\Gamma(\cdot)$ is increasing, exactly one of two subcases holds:

1. $g_j^*$ is such that $\Gamma(g_j^*) = \mathbf{x}_j^\top \sigma$, in which case the minimand (15) is $\ell_+(g_j^*) + \ell_-(g_j^*)$
2. $g_j^* = 1$ so that $\Gamma(g_j^*) = \Gamma(1) < \mathbf{x}_j^\top \sigma$, in which case the minimand (15) is $2\ell_+(1) + \mathbf{x}_j^\top \sigma$

A precisely analogous argument holds if $g_j$ falls in the second case, where $\mathbf{x}_j^\top \sigma < \Gamma(g_j)$. Putting the cases together, we have shown the form of the summand $\Psi$, piecewise over its domain, so (14) is equal to $\frac{1}{2}\min_{\sigma \geq \mathbf{0}^p}[\gamma(\sigma)]$.

We have proved the dependence of $g_j^*$ on $\mathbf{x}_j^\top \sigma^*$, where $\sigma^*$ is the minimizer of the outer minimization of (14). This completes the proof. $\qquad\square$

*Proof of Lemma 9.* We have

$$\max_{\substack{\mathbf{z}\in[-1,1]^n, \\ \mathbf{Fz}\geq n\mathbf{b}}} \frac{1}{n}\mathbf{z}^\top\mathbf{a} = \frac{1}{n}\max_{\mathbf{z}\in[-1,1]^n}\min_{\sigma\geq\mathbf{0}^p}\left[\mathbf{z}^\top\mathbf{a}+\sigma^\top(\mathbf{Fz}-n\mathbf{b})\right]$$

$$\overset{(a)}{=} \frac{1}{n}\min_{\sigma\geq\mathbf{0}^p}\max_{\mathbf{z}\in[-1,1]^n}\left[\mathbf{z}^\top(\mathbf{a}+\mathbf{F}^\top\sigma)-n\mathbf{b}^\top\sigma\right] \qquad (16)$$

$$= \frac{1}{n}\min_{\sigma\geq\mathbf{0}^p}\left[\left\|\mathbf{a}+\mathbf{F}^\top\sigma\right\|_1-n\mathbf{b}^\top\sigma\right] = \min_{\sigma\geq\mathbf{0}^p}\left[-\mathbf{b}^\top\sigma+\frac{1}{n}\left\|\mathbf{F}^\top\sigma+\mathbf{a}\right\|_1\right] \qquad (17)$$

where $(a)$ is by the minimax theorem ([20]). $\qquad\square$

## C.2 Other Proofs

**Lemma 10.** *The function $\ell_+(\Gamma^{-1}(m))+\ell_-(\Gamma^{-1}(m))$ is convex for $m \in (\Gamma(-1),\Gamma(1))$ under any of the conditions of Lemma 2.*

*Proof of Lemma 10.* Define $F(m) = \ell_+(\Gamma^{-1}(m)) + \ell_-(\Gamma^{-1}(m))$. By basic properties of the derivative,

$$\frac{d\left[\Gamma^{-1}\right]}{dm} = \frac{1}{\Gamma'(\Gamma^{-1}(m))} = \frac{1}{\ell'_-(\Gamma^{-1}(m)) - \ell'_+(\Gamma^{-1}(m))} \geq 0 \qquad (18)$$

where the last inequality follows by Assumption 1. Therefore, by the chain rule and (18),

$$F'(m) = \frac{\ell'_-(\Gamma^{-1}(m)) + \ell'_+(\Gamma^{-1}(m))}{\ell'_-(\Gamma^{-1}(m)) - \ell'_+(\Gamma^{-1}(m))} \qquad (19)$$

From this, we calculate $F''(m)$, writing $\ell'_\pm(\Gamma^{-1}(m))$ and $\ell''_\pm(\Gamma^{-1}(m))$ as simply $\ell'_\pm$ and $\ell''_\pm$ for clarity:

$$F''(m) = \underbrace{\frac{\frac{d[\Gamma^{-1}]}{dm}}{\left(\ell'_-(\Gamma^{-1}(m)) - \ell'_+(\Gamma^{-1}(m))\right)^2}}_{(a)}\left[\left(\ell'_- - \ell'_+\right)\left(\ell''_- + \ell''_+\right) - \left(\ell'_- + \ell'_+\right)\left(\ell''_- - \ell''_+\right)\right]$$

From (18), observe that the term $(a) = \left(\ell'_-(\Gamma^{-1}(m)) - \ell'_+(\Gamma^{-1}(m))\right)^{-3} \geq 0$. Therefore, it suffices to show that the other term is $\geq 0$. But this is equal to

$$\left(\ell'_- - \ell'_+\right)\left(\ell''_- + \ell''_+\right) - \left(\ell'_- + \ell'_+\right)\left(\ell''_- - \ell''_+\right) = 2(\ell'_-\ell''_+ - \ell''_-\ell'_+) \qquad (20)$$

This proves that condition (C) of Lemma 2 is sufficient for convexity of $F$ (and necessary also, under Assumption 1 on the partial losses).

We now address the other conditions of Lemma 2. (A) implies (C), because by Assumption 1, $\ell'_-, \ell''_+, \ell''_-$ are $\geq 0$ and $\ell'_+ \leq 0$, so (20) is $\geq 0$ as desired.

Finally we prove that (B) implies (C). If $\ell$ is proper, then it is well known (e.g. Thm. 1 of [16], and [6]) that for all $x \in (-1,1)$,

$$\frac{\ell'_-(x)}{1+x} = -\frac{\ell'_+(x)}{1-x}$$

(This is a simple and direct consequence of stationary conditions from the properness definition.)

Define the function $w(x) = \frac{\ell'_-(x)}{1+x} = -\frac{\ell'_+(x)}{1-x}$; we drop the argument and write it and its derivative as $w$ and $w'$ for clarity. By direct computation,

$$
\begin{aligned}
\ell'_-\ell''_+ - \ell''_-\ell'_+ &= [(1+x)w\,(w+(x-1)w')] - [(w+(1+x)w')(x-1)w] \\
&= [(1+x)w^2 + (x^2-1)ww'] - [(x-1)w^2 + (x^2-1)ww'] = 2w^2 \geq 0
\end{aligned}
$$

so (C) is true as desired. $\qquad\square$

*Proof of Lemma 2.* Continuity follows by checking $\Psi(m)$ at $m = \pm 1$.

For Lipschitzness, note that for $m \in (\Gamma(-1), \Gamma(1))$, by (19),

$$
\Psi'(m) = \frac{\ell'_-(\Gamma^{-1}(m)) + \ell'_+(\Gamma^{-1}(m))}{\ell'_-(\Gamma^{-1}(m)) - \ell'_+(\Gamma^{-1}(m))} \tag{21}
$$

$$
= -1 + \frac{2\ell'_-(\Gamma^{-1}(m))}{\ell'_-(\Gamma^{-1}(m)) - \ell'_+(\Gamma^{-1}(m))} \tag{22}
$$

$$
= 1 - \frac{2(-\ell'_+(\Gamma^{-1}(m)))}{\ell'_-(\Gamma^{-1}(m)) - \ell'_+(\Gamma^{-1}(m))} \tag{23}
$$

Using Assumption 1 on the partial losses, equations (22) and (23) respectively make clear that $\Psi'(m) \geq -1$ and $\Psi'(m) \leq 1$ on this interval. Since $\Psi'(m)$ is $-1$ for $m < \Gamma(-1)$ and $1$ for $m > \Gamma(1)$, it is 1-Lipschitz.

As for convexity, since $\Psi$ is linear outside the interval $(\Gamma(-1), \Gamma(1))$, it suffices to show that $\Psi(m)$ is convex inside this interval, which is shown in Lemma 10. $\qquad\square$

### C.3 Results and Proofs from Section 3

*Proof of Theorem 6.* The proof is similar to that of Theorem 4, which it generalizes. First note that

$$
\max_{\substack{\mathbf{z} \in [-1,1]^n, \\ \frac{1}{n}\mathbf{Fz} \geq \mathbf{b}}} \frac{1}{n} \sum_{i=1}^{n} r_j z_j \left(\ell_+(g_j) - \ell_-(g_j)\right) = \max_{\substack{\mathbf{z} \in [-1,1]^n, \\ \frac{1}{n}\mathbf{Fz} \geq \mathbf{b}}} -\frac{1}{n}\mathbf{z}^\top [\mathbf{r} \circ \Gamma(\mathbf{g})]
$$

$$
= \min_{\sigma \geq \mathbf{0}^p} \left[ -\mathbf{b}^\top \sigma + \frac{1}{n} \left\| \mathbf{F}^\top \sigma - (\mathbf{r} \circ \Gamma(\mathbf{g})) \right\|_1 \right] \tag{24}
$$

where the last equality uses Lemma 9.

Therefore, using (24) on the left-hand side of what we wish to prove,

$$
V = \frac{1}{2} \min_{\mathbf{g} \in [-1,1]^n} \left[ \frac{1}{n} \sum_{j=1}^{n} r_j \left[ \ell_+(g_j) + \ell_-(g_j) \right] + \max_{\substack{\mathbf{z} \in [-1,1]^n, \\ \frac{1}{n}\mathbf{Fz} \geq \mathbf{b}}} \frac{1}{n} \sum_{i=1}^{n} r_j z_j \left(\ell_+(g_j) - \ell_-(g_j)\right) \right]
$$

$$
= \frac{1}{2} \min_{\mathbf{g} \in [-1,1]^n} \left[ \frac{1}{n} \sum_{j=1}^{n} r_j \left[ \ell_+(g_j) + \ell_-(g_j) \right] + \min_{\sigma \geq \mathbf{0}^p} \left[ -\mathbf{b}^\top \sigma + \frac{1}{n} \sum_{j=1}^{n} \left| \mathbf{x}_j^\top \sigma - r_j \Gamma(g_j) \right| \right] \right]
$$

$$
= \frac{1}{2} \min_{\sigma \geq \mathbf{0}^p} \left[ -\mathbf{b}^\top \sigma + \frac{1}{n} \sum_{j=1}^{n} \min_{g_j \in [-1,1]} \left( r_j \left[ \ell_+(g_j) + \ell_-(g_j) \right] + \left| \mathbf{x}_j^\top \sigma - r_j \Gamma(g_j) \right| \right) \right] \tag{25}
$$

As in the proof of Theorem 4, the inner minimization's objective can be simplified:

$$
r_j(\ell_+(g_j) + \ell_-(g_j)) + \left| \mathbf{x}_j^\top \sigma - r_j \Gamma(g_j) \right| = \begin{cases} 2r_j\ell_+(g_j) + \mathbf{x}_j^\top \sigma & \text{if } \mathbf{x}_j^\top \sigma \geq r_j \Gamma(g_j) \\ 2r_j\ell_-(g_j) - \mathbf{x}_j^\top \sigma & \text{if } \mathbf{x}_j^\top \sigma < r_j \Gamma(g_j) \end{cases} \tag{26}
$$

Suppose $g_j$ falls in the first case, so that $\mathbf{x}_j^\top \sigma \geq r_j \Gamma(g_j)$. From Assumption 1, $2r_j\ell_+(g_j) + \mathbf{x}_j^\top \sigma$ is decreasing in $g_j$, so it is minimized for the greatest $g_j^* \leq 1$ s.t. $\Gamma(g_j^*) \leq \frac{\mathbf{x}_j^\top \sigma}{r_j}$. Since $\Gamma(\cdot)$ is increasing, exactly one of two subcases holds:

a) $g_j^*$ is such that $\Gamma(g_j^*) = \frac{\mathbf{x}_j^\top \sigma}{r_j}$, in which case the minimand (26) is $r_j(\ell_+(g_j^*) + \ell_-(g_j^*))$

b) $g_j^* = 1$ so that $\Gamma(g_j^*) = \Gamma(1) < \frac{\mathbf{x}_j^\top \sigma}{r_j}$, in which case the minimand (26) is $2r_j\ell_+(1) + \mathbf{x}_j^\top \sigma$

A precisely analogous argument holds if $g_j$ falls in the second case, where $\mathbf{x}_j^\top \sigma < \Gamma(g_j)$. So as before, we have proved the dependence of $g_j^*$ on $\mathbf{x}_j^\top \sigma^*$, where $\sigma^*$ is the minimizer of the outer minimization of (25). This completes the proof. $\qquad\square$

**Theorem 11.** *For any* $\mathbf{c} \geq \mathbf{0}^n$,

$$
\min_{\mathbf{g} \in [-1,1]^n} \ \max_{\substack{\mathbf{z} \in [-1,1]^n, \\ \left|\frac{1}{n}\mathbf{Fz} - \mathbf{b}\right| \leq \mathbf{c}}} \ \ell(\mathbf{z}, \mathbf{g}) \ = \ \min_{\sigma \in \mathbb{R}^p} \left[ -\mathbf{b}^\top \sigma + \frac{1}{n}\sum_{j=1}^n \Psi(\mathbf{x}_j^\top \sigma) + \mathbf{c}^\top |\sigma| \right]
$$

*Writing* $\sigma_\mathbf{c}^*$ *as the minimizer of the RHS above, the optimal predictions* $\mathbf{g}^* = \mathbf{g}(\sigma_\mathbf{c}^*)$*, as in Theorem 4.*

*Proof of Theorem 11.* This is proved exactly like Theorem 4, but using Lemma 12 instead of Lemma 9 in that proof. $\qquad\square$

**Lemma 12.** *For any* $\mathbf{a} \in \mathbb{R}^n$ *and* $\mathbf{c} \geq \mathbf{0}^n$,

$$
\max_{\substack{\mathbf{z} \in [-1,1]^n, \\ \left|\frac{1}{n}\mathbf{Fz} - \mathbf{b}\right| \leq \mathbf{c}}} \frac{1}{n}\mathbf{z}^\top \mathbf{a} \ = \ \min_{\sigma \in \mathbb{R}^p} \left[ -\mathbf{b}^\top \sigma + \frac{1}{n} \left\| \mathbf{F}^\top \sigma + \mathbf{a} \right\|_1 + \mathbf{c}^\top |\sigma| \right]
$$

*Proof.*

$$
\max_{\substack{\mathbf{z} \in [-1,1]^n, \\ \left|\frac{1}{n}\mathbf{Fz} - \mathbf{b}\right| \leq \mathbf{c}}} \frac{1}{n}\mathbf{z}^\top \mathbf{a} = \max_{\substack{\mathbf{z} \in [-1,1]^n, \\ \frac{1}{n}\mathbf{Fz} - \mathbf{b} \leq \mathbf{c}, \\ -\frac{1}{n}\mathbf{Fz} + \mathbf{b} \leq \mathbf{c}}} \frac{1}{n}\mathbf{z}^\top \mathbf{a}
$$

$$
= \frac{1}{n} \max_{\mathbf{z} \in [-1,1]^n} \min_{\lambda, \xi \geq \mathbf{0}^p} \left[ \mathbf{z}^\top \mathbf{a} + \lambda^\top(-\mathbf{Fz} + n\mathbf{b} + n\mathbf{c}) + \xi^\top(\mathbf{Fz} - n\mathbf{b} + n\mathbf{c}) \right]
$$

$$
= \frac{1}{n} \min_{\lambda, \xi \geq \mathbf{0}^p} \max_{\mathbf{z} \in [-1,1]^n} \left[ \mathbf{z}^\top(\mathbf{a} + \mathbf{F}^\top(\xi - \lambda)) + \lambda^\top(n\mathbf{b} + n\mathbf{c}) + \xi^\top(-n\mathbf{b} + n\mathbf{c}) \right]
$$

$$
= \frac{1}{n} \min_{\lambda, \xi \geq \mathbf{0}^p} \left[ \left\| \mathbf{a} + \mathbf{F}^\top(\xi - \lambda) \right\|_1 - n\mathbf{b}^\top(\xi - \lambda) + n\mathbf{c}^\top(\xi + \lambda) \right]
$$

where the interchanging of min and max is again justified by the minimax theorem ([20]), since the objective is linear in each variable and one of the constraint sets is closed.

Suppose for some $j \in [n]$ that $\xi_j > 0$ and $\lambda_j > 0$. Then subtracting $\min(\xi_j, \lambda_j)$ from both does not affect the value $[\xi - \lambda]_j$, but always decreases $[\xi + \lambda]_j$, and therefore always decreases the objective function. Therefore, we can w.l.o.g. assume that $\forall j \in [n] : \min(\xi_j, \lambda_j) = 0$. Defining $\sigma_j = \xi_j - \lambda_j$ for all $j$ (so that $\xi_j = [\sigma_j]_+$ and $\lambda_j = [\sigma_j]_-$), the last equality above becomes

$$
\frac{1}{n} \min_{\lambda, \xi \geq \mathbf{0}^p} \left[ \left\| \mathbf{a} + \mathbf{F}^\top(\xi - \lambda) \right\|_1 - n\mathbf{b}^\top(\xi - \lambda) + n\mathbf{c}^\top(\xi + \lambda) \right]
$$

$$
= \frac{1}{n} \min_{\sigma \in \mathbb{R}^p} \left[ \left\| \mathbf{a} + \mathbf{F}^\top \sigma \right\|_1 - n\mathbf{b}^\top \sigma + n\mathbf{c}^\top |\sigma| \right]
$$

$\qquad\square$