[Reviews · NeurIPS 2016]

Reviewer 1

Summary

In this paper a worst-case optimal aggregation algorithm for binary classifiers under general losses is proposed. This work builds upon a line of work on semi-supervised and transductive learning for classification problems, most notably ref. [1]. The main contribution of the paper is a minimax formulation of the problem exploiting classifier correlation constraints. It is proved that this formulation leads to a convex program, giving a closed-form minimax optimal predictor under rather general conditions on the loss function (and even the type of side constraints used, but this is left in the Appendix). This result is based on a simple application of Sion minimax Theorem, together with a characterization of convexity of the minimax objective, the so-called the Potential Well. As said before, this paper builds upon existing work, but it extends the applicability of these ideas very broadly. Technically, the paper does not introduce heavy machinery, but the obtained insights lead to new understanding of the problem which seems to be absent in the literature so far. On top of this, it is a very well written paper. For these reasons, I recommend this paper for publication at NIPS.

Qualitative Assessment

I believe this paper has great potential impact. First, it tackles a central question in learning theory (semi-supervised classification), it generalizes the literature considerably, and the obtained tools are scalable and practical. The technical content of the paper is good, although not extremely novel; despite this, it is able to provide insights that were not previously obtained in the literature. A natural high-level question is whether the minimax approach is too conservative in a more benign setting. A possible way to formalize this question is whether the minimax results obtained can be translated to generalization bounds in the distributional setting, possibly by some additional tools such as regularization. Next I would like to make some minor comments: 1. I couldn't find the justification of the "perfectly tight" minimax bound (Line 97). However, this is evident from Sion minimax Theorem (this is a zero sum game, which is quasiconcave on one variable, and convex --in fact, linear-- in the other), and I suggest to add this as a reference. 2. The discussion on lines 229-233 is confusing. I quickly checked [1], and I couldn't understand where the data distribution is used. Please clarify. 3. Finally, the equation following line 288 needs some further justification.

Confidence in this Review

2-Confident (read it all; understood it all reasonably well)


Reviewer 2

Summary

Nice paper extending previous work on 0-1 loss ensemble aggregation to a much more general family of losses.

Qualitative Assessment

This paper tackles the problem of "aggregating" binary classifiers for semi-supervised learning for a general class of loss functions. This paper extends the ideas from 0-1 loss from a previous work and apply it in the case where the losses monotonically non-increase away from the decision boundary and are twice differentiable. This includes twice differentiable convex loss functions. This paper again writes this problem as a zero-sum game, sufficiently constraining the adversary. Using the dual formulation, they show the relevant algorithm minimizes the "slack function" and use the predictor g(\sigma). This has nice connections to many other algorithms. I had immediately though this should cover the AdaBoost loss function, and I was happy to see that mentioned in the table. My overall impression of this work is positive, though I have a couple of " complaints". First, it would be nice to know which contributions are really novel to this work, and which are novel to [1]. For example, it seems the minimax formulation is from [1], just with different losses. Second, I don't really see a deep connection here to semi-supervised learning, which is hardly mentioned anywhere but the title. This seems to just work for building an optimal ensemble from a set of classifiers. Some clarification why semi-supervised learning appears in the abstract, and almost nowhere else, would be useful.

Confidence in this Review

2-Confident (read it all; understood it all reasonably well)


Reviewer 3

Summary

This paper provides a family of parameter-free ensemble aggregation algorithms of binary classifiers which use labeled and unlabeled data; these are as efficient as linear learning and prediction for convex risk minimization, but work without any relaxations on many non-convex loss functions. Also, it proves the family of ensemble aggregation algorithms has a better expected test loss guarantee than any weighted majority under the same ensemble constraints.

Qualitative Assessment

The paper addresses the problem of aggregating an ensemble of binary classifiers in a semi-supervised setting, and generalizes the 0-1 loss in a previous work to a more general class of loss functions. The family of algorithms proposed can solve the problem perfectly and efficiently with some assumptions. The paper clearly explains the theory used and describes the methods with sufficient details. In addition, the assumptions are discussed and their effect on learning part of the algorithm is well explained. The paper is well organized except Chapter 4 from which Section 4.2 seems independent.

Confidence in this Review

3-Expert (read the paper in detail, know the area, quite certain of my opinion)


Reviewer 4

Summary

The authors consider the following problem. Input: a set of classifiers h_1,...,h_p and a set of unlabeled data x_1,..., x_n such that h_j(x_i) \in {-1,1}.x Output: a prediction in [-1,1] for each x_i that minimizes the following loss function: l(z_j,g_j) = (1+z_j)(1-g_j)/4 + (1-z_j)(1+g_j)/4, where z_j corresponds to the correct label (in [-1,1]) of x_j. This problem is a generalization of a recent work by [1] where they only focused in loss function in {0,1} (and not [0,1]). The authors formulate the problem as a 2-player game: one player being the predictor, the other choosing the values of the z_is, based on the constraints induced by the classifiers h_1,...,h_p. This formulation translates into a convex optimization problem whose solution is shown to be optimal. Lemma 2 combined with standard optimization techniques allow to get an optimal solution. Then, for each element x_i, the algorithm predicts its label by applying the solution function obtained for the optimization problem on x_1,...,x_{i-1}. They show that the optimal prediction for x_i is a sigmoid-like function of a linear combination of the predictions h_j(x_i).

Qualitative Assessment

I find the paper quite well-written and I think that this line of work is very interesting: the original problem is quite fundamental. However, it is not clear to me how interesting the generalization of the original problem (that was considered in [1]) is (it is stated that "the zero-one loss is inappropriate for other common binary classification tasks, such as estimating label probabilities, and handling false positives and false negatives differently." but some concrete examples and references would be interesting). Moreover, I am not sure how novel the techniques are compared [1] since the reformulation was already present.

Confidence in this Review

2-Confident (read it all; understood it all reasonably well)


Reviewer 5

Summary

This paper considers the problem of optimally aggregation of learned classifiers for improving the performance on unlabelled test data. The work is the generalisation of [1], a COLT15 paper. The contribution lies on generalising the analysis from 0-1 loss to broader class of general losses.

Qualitative Assessment

+ The paper considers a very interesting topic which relates ensemble methods to semi-supervised learning. Comparing to [1], the work generalises the analysis to broader class of losses, which is meaningful in practice. The theoretical results are sound. - > From the analysis technique and the main results, it seems that this work is incremental to [1], which limits the originality and contributions. > One crucial assumption in the paper is that the performance of classifiers, denoted as "b", can be estimated with sufficient accuracy using labeled data. While in common semi-supervised learning, the number of labeled data is quite limited, so estimation of classifier performance is usually difficult. This may cause the algorithm impractical to use.

Confidence in this Review

2-Confident (read it all; understood it all reasonably well)